# Prediction of Blast-Induced Ground Vibration Using an Adaptive Network-Based Fuzzy Inference System

Primož Jelušič [ORCID], Andrej Ivanič and Samo Lubej *

Faculty of Faculty of Civil Engineering, Transportation Engineering and Architecture, University of Maribor, Smetanova ulica 17, 2000 Maribor, Slovenia; primoz.jelusic@um.si (P.J.); andrej.ivanic@um.si (A.I.)
* Correspondence: samo.lubej@um.si; Tel.: +386-(2)-22-94-333

**Abstract:** Efforts were made to predict and evaluate blast-induced ground vibrations and frequencies using an adaptive network-based fuzzy inference system (ANFIS), which has a fast-learning capability and the ability to capture the non-linear response during the blasting process. For this purpose, the ground vibrations generated by the blast in a tunnel tube were monitored at a residential building located directly above the tunnel tube. To investigate the usefulness of this approach, the prediction by the ANFIS was also compared to those by three of the most commonly used vibration predictors. The efficiency criteria chosen for the comparison between the predicted and actual data were the sum of squares due to error (SSE), the root mean squared error (RMSE), and the goodness of fit (R-squared and adjusted R-squared). The results show that the ANFIS prediction model performs better than the commonly used predictors.

**Keywords:** peak particle velocity; frequency; predictor equation; adaptive network-based fuzzy inference system

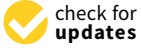



## 1. Introduction

An increasing number of tunnel construction sites are located under or near inhabited buildings. Since houses, hospitals, and other buildings often contain sensitive equipment, special efforts must be made to reduce vibrations during blasting operations. The relationship between the peak level of the vibration, the distance from the source to the monitor and the total charge weight in the blast hole is defined by the charge weight scaling law [1]. Therefore, buildings and installations within a certain radius should be inspected and given a value for the peak particle velocity (PPV) level based on the type of ground underlying the building, its foundation, material content, and type of construction. The maximum vibration level for a building built on rock is much higher than for that built on clay, because the vibration frequency in rock is much higher than in clay, which increases the probability that the vibration will cause damage to the buildings. The vibration level during a blast depends mainly on the transfer conditions of the ground, the amount of charge, and the distance between the explosive charge and the measuring points.

The monitoring of vibrations generated by blasting is an important component in the control of environmental stability and blast effectiveness. Various mathematical and empirical formulas have been developed to predict blast-induced ground vibrations. These empirical equations usually show high deviations between the measured and calculated peak particle velocity (PPV) values.

Many engineers and researchers have discovered that artificial neural networks (ANNs) and the neuro-fuzzy approach have better predictive capabilities than conventional vibration predictors. Khandelwal and Singh [2] used ANN to predict the blast-induced ground vibration level at a magnesite mine based on 75 blast events. Later, Khandelwal and Singh [3] developed an ANN model for predicting ground vibrations and frequencies that considers not only the distance from the blast site and the charge per delay, but also rock properties, blast design, and explosive parameters. The sensitivity analysis of

different types of ANN models showed that the distance from the blast site, the number of boreholes per delay, and the maximum charge per delay are the most effective parameters in generating ground vibrations during blasting [4,5]. Kostič et al. developed a neural network model with four main blast parameters as input, namely total charge, maximum charge per delay, distance from the blast source to the measuring point and hole depth [6]. To evaluate the ground vibrations caused by the blast, the techniques of dimensional analysis and ANN were applied, taking into account the blast design parameters and rock strength [7]. The control of the blast induced vibrations during the construction of the Masjed Soleiman dam was the crucial task, therefore the general regression neural network (GRNN) and the support vector machine (SVM) were used to predict the vibrations [8]. A hybrid model of the ANN and a particle swarm optimization (PSO) algorithm was implemented to predict ground vibrations based on 88 blast events [9]. Two novel hybrid artificial intelligent models for predicting the blast-induced peak particle velocity were presented by Li et al. [10]. Rao and Rao [11] applied the neuro-fuzzy technique for ground vibration and frequency prediction in opencast mine while Khendelwal [12] applied the ANN. Suchatvee et al. [13] investigated the advantages and limitations of ANNs for the prediction of surface settlements generated by earth pressure balance shield tunneling. The importance of site-specific factors in the prediction model for blast-induced ground vibrations was presented by Kuzu [14]. ANNs have also been used to analyze the surface settlements caused by shotcrete-supported tunnels [15] and to predict the Tunnel Boring Machine Penetration Rate [16]. The summary of the previous studies on the PPV, which include several soft computing and machine learning methods, was presented by Zhang et al. [17].

In this study, an attempt was made to predict and evaluate blast-induced ground vibrations and frequencies by integrating the maximum charge per delay (maximum quantity of explosive charge detonated on one interval within a blast) and the distance between the blast surface and the vibration monitoring point using an adaptive network-based fuzzy inference system (ANFIS). The modeled ground vibrations were generated by blasting in a tunnel tube. Iphar et al. [18] also applied the ANFIS model for the PPV prediction, but for the vibrations generated by the blast in an open-pit mining. To investigate the usefulness of this approach, the prediction by the ANFIS was compared to commonly used vibration predictors. To ensure that the comparison between the conventional predictors and the ANFIS model was realistic, the ANFIS model was designed with only two inputs, as per conventional predictors.

## 2. Mechanism of Ground Vibration and the Vibration of Buildings

Stress waves generated by blasting cause a displacement of the ground, which is expressed in the form of periodic and non-stationary fluctuations. When stress waves hit a building, some of the energy in the ground is transferred to the building's foundation. The concentration of vibrations can lead to permanent foundation settlement and partial failure of the building's construction elements. Vibrations are seismic waves that travel through a material, and seismic waves caused by blasting are usually referred to as vibrations, shocks, or blast-induced vibrations. There are three types of seismic waves generated by blasting:

- Pressure waves that produce oscillating compressive stress and shear stress—these stresses are propagated in the wave direction. In rock masses, pressure waves propagate both through the mineral structure and through the pores, which is why the pressure wave velocity is increased in a saturated hard rock.
- Shear waves oscillating perpendicular to the wave direction propagate only through the mineral structure, therefore, water saturation only has a small influence on the velocity of the shear waves. The energy of shear waves is less easily transmitted through rock mass in comparison to the energy of primary waves.
- Surface waves.

The result of blasting depends more on the properties of the rock than on the explosives used to break the rock. These properties are tensile and compressive strength, density,

and seismic velocity. High-density rock is generally more difficult to blast than lower-density rock. Cracks, fissures, and other weak zones also influence seismic waves. The properties of various types of rock are shown in Table 1 [19]. However, these properties apply to intact rock, which must be reduced due to the properties of joints in a rock mass. Therefore, several empirical equations have been developed to estimate the value of an isotropic rock mass deformation modulus. The most important classification systems for the characterization of rock mass are the Geological Strength Index (GSI) [20], the Rock Mass Rating (RMR) [21], and the Tunneling Quality Index (Q) [22]. Several studies have related pressure wave velocity ($V_p$) with mechanical and physical properties such as stiffness, strength and density of rock mass [23]. The $V_p$ for elastic and isotropic media is calculated with Equation (1) [24]:

$$V_p = \sqrt{\frac{K_b + \frac{4}{3} \cdot G}{\rho}} \tag{1}$$

where $K_b$ is the bulk modulus, $G$ is the shear modulus, and $\rho$ is the bulk density. The summary of relationship between pressure wave velocity with density and uniaxial compressive strength was presented by Yagiz [25].

**Table 1.** Rock characteristics influencing blasting.

| Type of Rock | Density (kg/m$^3$) | Seismic Velocity (m/s) | Compressive Strength (MPa) | Tensile Strength (MPa) |
|---|---|---|---|---|
| Granite | 2700–2800 | 4500–6000 | 200–360 | 10–30 |
| Diabase | 2800–3100 | 4000–5000 | 290–400 | 19–30 |
| Limestone | 2400–2700 | 3000–4500 | 130–200 | 17–30 |
| Marble | 2800–3000 | 6000–7000 | 150–190 | 15–25 |

Blasting is a demanding and exact form of engineering, which is why proper planning is required. This includes the following:

- Choosing the depth, spacing, and geometry of blast holes;
- Choosing the type and optimal amount of explosives;
- Choosing the maximum quantity of charge per initiation interval.

In the vicinity of listed and historic buildings, any necessary blasting should be carried out by a series of small chain explosions. In this way, the effects of vibrations on buildings can be kept to a minimum. If the blasting procedure is correctly designed, most of the energy will be absorbed in the blast field.

Since there are several ways in which a building can vibrate due to the effect of surface waves, it is desirable that the vibration components in the three principal directions are measured simultaneously in order to make the accurate assessment possible. The concept of maximum resultant motion is widely used.

The information on the vibration levels required for damage is contained in several standards, such as the standard DIN 4150 Vibrations in Buildings [26], the SN 640 312a standard on the vibration effects on buildings [27], and the USBM RI 8507 code/guideline/recommendations. Among these standards, DIN 4150 is the most conservative and restrictive, and aims to reduce vibration effects and complaints [28]. Standard DIN 4150 also specifies safe vibration levels according to the type and condition of a building; thus, allowance is made for the type of building. The Norwegian Standard [29] limits the PPV to 80 mm/s for a structure made of reinforced concrete founded directly on hard rock. The limit values of vibrations, which take into account the magnitude, frequency, duration of the vibration, and the type of building are also provided in the British Standard BS 7385-2 [30]. The guideline values of the vibrations of various standards were summarized by Norén-Cosgriff et al. [31].

Vibration standards are typically plotted graphically using logarithmic scales in both vertical and horizontal directions. The peak particle velocity (m/s) is presented on the vertical scale and the vibration frequency (Hz) is on the horizontal scale, as shown in Figure 1 [28].

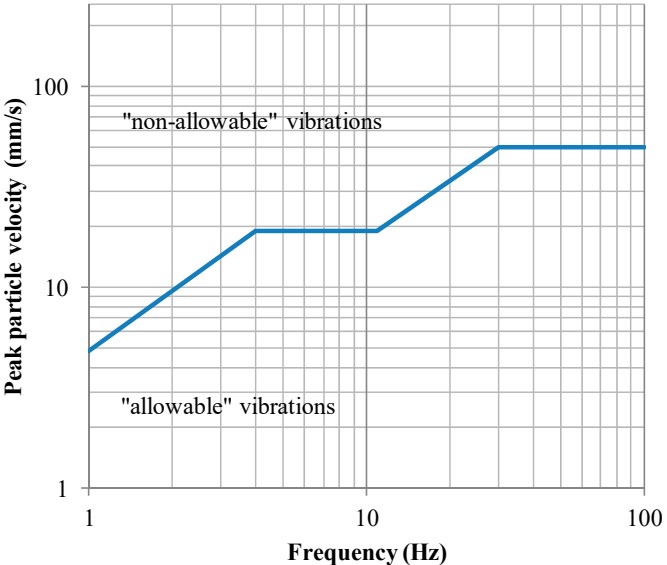

**Figure 1.** Safe levels of blasting vibrations for houses.

Ground vibrations caused by blasting in tunnel construction can cause structural failures. Therefore, the effects of blast vibrations on buildings must be reduced or suppressed. The intensity of ground vibrations depends on controllable and non-controllable parameters. The main controllable parameters are the type and quantity of the explosive, sequence of initiation, powder factor, drilling, steaming, and hole depth while the main uncontrollable parameters are rock mass properties, geology, and joint formation. The vibration level at a certain point away from the charge is proportional to the weight of the explosive, while it is inversely proportional to the distance. Equations that are currently used for calculating and estimating the vibration level and the amount of charge per delay generally show high deviations between the measured and calculated peak particle velocity (PPV) values.

The following conventional empirical equation provides the intensity of vibration in the form of PPV [19]:

$$PPV = K \cdot \left( \frac{Q^{\alpha}}{d^{\beta}} \right) \tag{2}$$

where *PPV* (mm/s) is the peak particle velocity, *K* (-) is the rock constant, *Q* (kg) is the charge per detonator delay number, *d* (m) is the distance between the blasting and measurement points, and $\alpha$ (-) and $\beta$ (-) are the damping coefficients. As the damping coefficients are difficult to determine, the equation does not provide sufficient results [19]. To determine the damping coefficient, the measured attenuation data should be well-matched with the predicted data [32]. For practical reasons, several authors have proposed simplified equations, which are listed in Table 2. The on-site constants (*K*, *B*) can be obtained by using multiple regression analysis. The conventional blast vibration estimators are not able to estimate the *PPV* up to an acceptable limit because of the data scattering of blast vibrations. Agrawal and Mishra [33] reported that the errors between predicted and actual PPV are due to the fact that cap scattering in the pyrotechnic based delay initiation system varies between ± 10% and ± 20%.

**Table 2.** Different vibration predictor equations.

| Equations Named After Authors | Equations |
|:---:|:---:|
| USBM [34] | $PPV = K\left(\frac{d}{\sqrt{Q_{MAX}}}\right)^{-B}$ |
| Ambraseys–Hendron [35] | $PPV = K\left(\frac{d}{Q_{MAX}^{1/3}}\right)^{-B}$ |
| Langefors–Kihlstorm [36] | $PPV = K\left(\sqrt{\frac{Q_{MAX}}{d^{2/3}}}\right)^{B}$ |

## 3. Tunnel Site and Measurements

During the construction of the 7.6 km long motorway section Pluska–Ponikve (Slovenia) two tunnels had to be built. Above the tunnel tube was a house, which was exposed to vibrations due to the blasting (Figure 2). Therefore, the vibration levels were monitored. The house had a basement and two upper floors. The load-bearing walls of the house were built using stone and lime mortar with an estimated compressive strength of 0.5 MPa. The floors were built from reinforced concrete slabs. The facility where the monitoring was performed is shown in Figure 3.

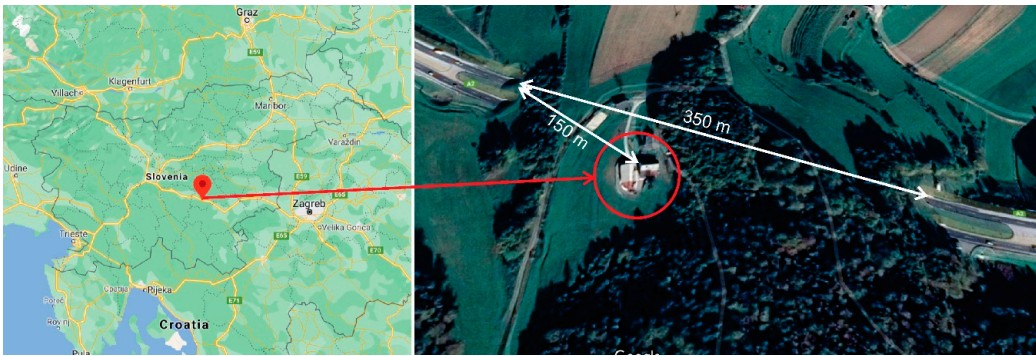

**Figure 2.** Tunnel tube map—highway Pluska–Ponikve.

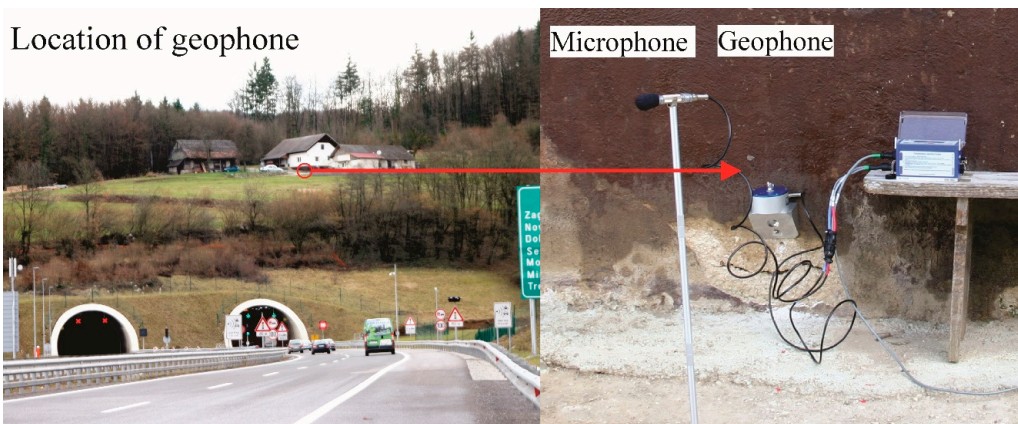

**Figure 3.** Facility where the monitoring was performed.

The measurements of vibrations induced by blasting in the tunnel tubes were obtained using Minimate Plus (Instantel) measuring equipment with a triaxial geophone, which was placed on the load-bearing wall of the building at the height of 0.6 m above the ground. A continuous record mode was used to record multiple events automatically with no dead time between blast events. The geophone records a blast event and then continues to monitor, ready to record the following events. The geophone records all blast events with a PPV exceeding the 0.2 mm/s. The blasting in two tunnel tubes was carried out from March to the end of July 2008. During this period, there were 48 measurements of blasting

events. On the wall of the building, the PPV was measured in the longitudinal, vertical, and transverse directions. For each blasting event, the vibration frequency, amplitude, and peak particle velocity were recorded, as can be seen in Figure 4. The deviations of the fundamental natural frequencies can influence the structural strength of the building. However, for failure mechanisms such as bending and shear, where the building is forced to follow the oscillatory movements of the ground surface, the deviation of the fundamental natural frequencies of the building is of minor importance [31]. Table 3 presents the measurements of 40 of the blasting events that were used to define the site constants $K$ and $B$ (training data), while Table 4 contains the remaining eight measurements that were used to evaluate the prediction capability of the conventional vibration predictors (testing data). Before the blasting of the tunnel tubes, the 7 cracks were found on the outer walls of the building (see, Figure 5). We mapped the cracks and measured their width. The movements or displacements of these cracks were observed with installed plaster seals. During the blasting, it was revealed that the cracks did not enlarge, and no new cracks formed.

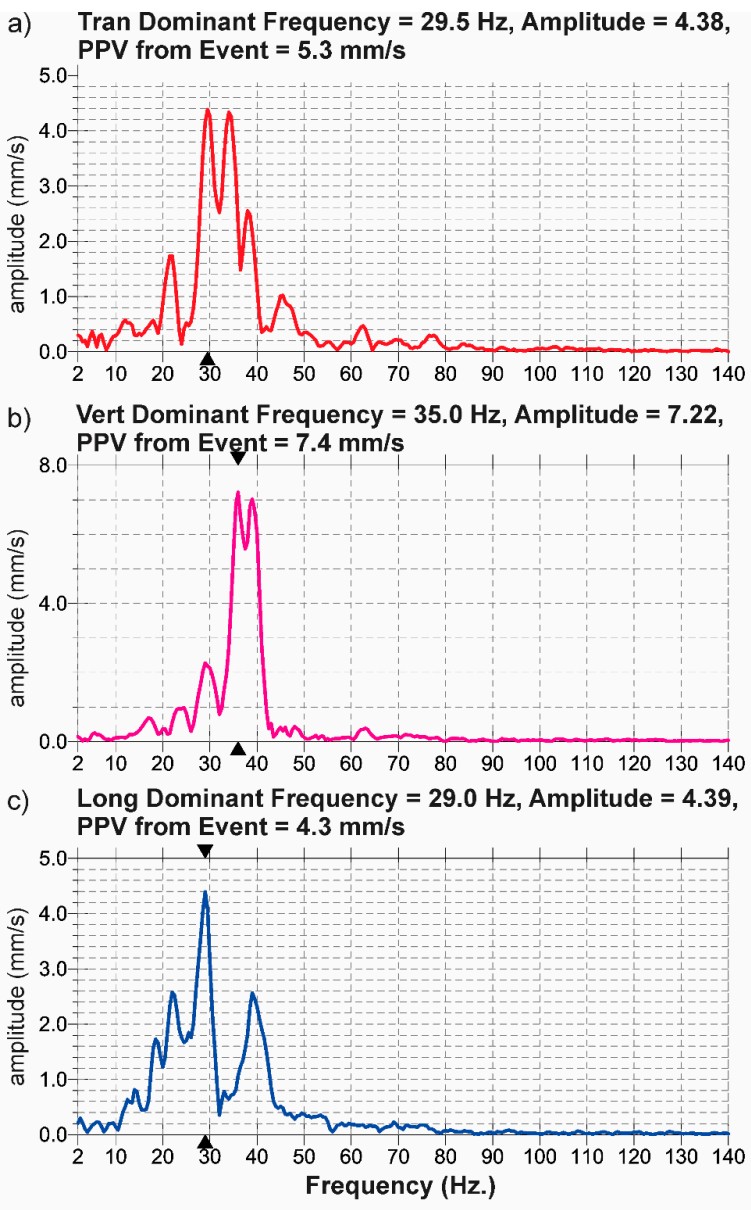

**Figure 4.** Records of vibrations caused by blasting event. (**a**) transverse vibrations, (**b**) vertical vibrations and (**c**) longitudinal vibrations.

**Table 3.** Training data.

| Measurement No. | Explosive (kg) | Distance (m) | PPV (mm/s) | Frequency (Hz) |
|---|---|---|---|---|
| 1 | 32 | 74 | 4.1 | 74 |
| 2 | 33 | 80 | 5.5 | 80 |
| 3 | 35 | 95 | 2.9 | 95 |
| 4 | 35 | 90 | 2.7 | 90 |
| 5 | 37 | 82 | 3.6 | 82 |
| 6 | 38 | 98 | 2.6 | 98 |
| 7 | 39 | 59 | 6.0 | 59 |
| 8 | 41 | 33 | 4.7 | 33 |
| 9 | 41 | 70 | 6.4 | 70 |
| 10 | 42 | 62 | 6.5 | 62 |
| 11 | 43 | 89 | 2.7 | 89 |
| 12 | 44 | 35 | 7.4 | 35 |
| 13 | 44 | 94 | 2.1 | 94 |
| 14 | 44 | 83 | 2.7 | 83 |
| 15 | 45 | 68 | 5.8 | 68 |
| 16 | 45 | 86 | 3.4 | 86 |
| 17 | 47 | 108 | 1.9 | 108 |
| 18 | 47 | 92 | 2.6 | 92 |
| 19 | 47 | 103 | 2.5 | 103 |
| 20 | 47 | 70 | 4.1 | 70 |
| 21 | 49 | 44 | 7.5 | 57 |
| 22 | 49 | 72 | 5.5 | 72 |
| 23 | 49 | 62 | 4.4 | 62 |
| 24 | 49 | 66 | 7.2 | 66 |
| 25 | 49 | 61 | 1.7 | 61 |
| 26 | 49 | 102 | 3.1 | 102 |
| 27 | 50 | 56 | 7.8 | 56 |
| 28 | 50 | 32 | 17.5 | 32 |
| 29 | 50 | 87 | 2.4 | 39 |
| 30 | 50 | 91 | 3.6 | 91 |
| 31 | 51 | 96 | 2.4 | 37 |
| 32 | 53 | 65 | 8.8 | 65 |
| 33 | 53 | 101 | 3.0 | 101 |
| 34 | 53 | 32 | 8.1 | 32 |
| 35 | 54 | 76 | 4.9 | 76 |
| 36 | 55 | 79 | 5.7 | 79 |
| 37 | 56 | 64 | 6.0 | 64 |
| 38 | 56 | 39 | 10.9 | 39 |
| 39 | 62 | 65 | 10.0 | 65 |
| 40 | 63 | 32 | 12.7 | 32 |

**Table 4.** Testing data.

| Measurement No. | Explosive (kg) | Distance (m) | PPV (mm/s) | Frequency (Hz) |
|---|---|---|---|---|
| 1 | 33 | 104 | 2.2 | 43 |
| 2 | 36 | 63 | 7.3 | 57 |
| 3 | 40 | 106 | 2.8 | 47 |
| 4 | 42 | 71 | 4.9 | 73 |
| 5 | 44 | 58 | 5.5 | 73 |
| 6 | 46 | 32 | 7.9 | 85 |
| 7 | 47 | 84 | 3.7 | 51 |
| 8 | 54 | 63 | 7.7 | 85 |

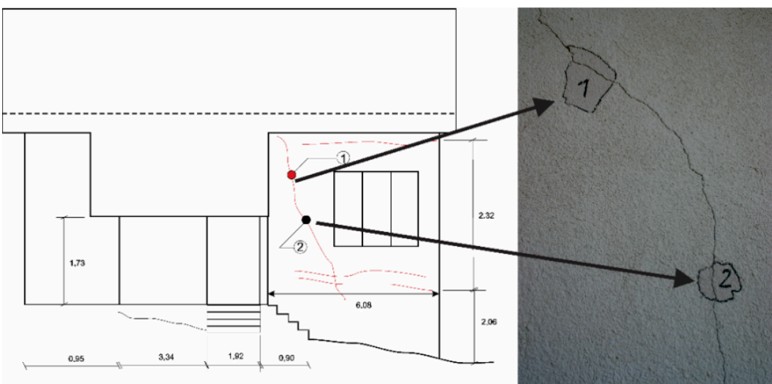

**Figure 5.** Measurements of the crack width on the building.

## 4. Conventional Vibration Predictors

To estimate the site coefficients *K* and *B* of each conventional vibration predictor, the least squares method was used, which minimizes the summed square of the residual value. The least squares method is often used to generate estimators and other statistics in regression analysis. The residual value for the $i$th data point $r_i$ is designated as the difference between the fitted response value $x'_i$ and the observed response value $x_i$, and is described as the fitting data error, see Equation (3).

$$r_i = x_i - x'_i \tag{3}$$

The summed square of the residual value is calculated by

$$Sum = \sum_{i=1}^{n} r_i^2 = \sum_{i=1}^{n} (x_i - x\prime_i)^2 \tag{4}$$

where $n$ is the number of data points contained in the fitting process and the sum of squares due to error is denoted as *Sum*.

The site coefficients were determined based on the training data for three different conventional predictors proposed by USBM [34], Ambraseys and Hendron [35], and Langefors and Kihlstorm [36]. The USBM [34] predictor is used while it provides the safe level blasting criteria, although it is used to predict blast induced ground vibrations from surface mining rather than tunnel blasting. In order to evaluate the quality of the fit, four statistical parameters were evaluated: R-squared, sum of squares due to error (SSE), root mean squared error (RMSE), and adjusted R-squared. SSE calculates the total difference of the response values from the fit to the response values. A value near zero indicates that the model has a lower random error, and that the fit will be more appropriate for prediction. R-squared is the square of the correlation between the predicted response values and the actual response values. R-squared can take the value of any number between zero and one, with a value near one demonstrating that a larger proportion of variance is considered. RMSE estimates the standard deviation of the random component in the data. An RMSE value near zero indicates a fit that is more appropriate for prediction. The estimated site coefficients of each prediction model along with the goodness of fit are presented in Table 5.

**Table 5.** Estimated site coefficients based on the training data and goodness of fit of each prediction equation.

| Name of predictor | Equation | Site Coefficient | | SSE (mm/s) | Goodness of Fit | | RMSE (mm/s) |
| | | B | K | | R-Squared | Adjusted R-Squared | |
|---|---|---|---|---|---|---|---|
| USBM [34] | $PPV = K\left(\dfrac{d}{\sqrt{Q_{MAX}}}\right)^{-B}$ | 1.033 | 53.18 | 159.12 | 0.62 | 0.61 | 2.05 |
| Ambraseys–Hendron [35] | $PPV = K\left(\dfrac{d}{Q_{MAX}^{1/3}}\right)^{-B}$ | 1.047 | 107.7 | 165.1 | 0.61 | 0.60 | 2.08 |
| Langefors–Kihlstorm [36] | $PPV = K\left(\sqrt{\dfrac{Q_{MAX}}{d^{2/3}}}\right)^{B}$ | 2.652 | 1.255 | 151.49 | 0.64 | 0.63 | 2.00 |

Note: SSE, sum of squares due to error; RMSE, root mean square error.

## 5. ANFIS Model for Vibration Prediction Called ANFISBLAST

The main structure of the fuzzy inference system (FIS) was developed by Zadeh [37]. In this FIS basic structure, it is necessary to choose the type and number of membership functions. Membership functions, which are determined by humans, are usually subjective and depend on individual preferences. Precisely defined methods to translate human experience and human knowledge into membership functions and fuzzy rules do not exist. Often, there is a set of input/output data on which to build an FIS model. The adaptive network-based FIS (ANFIS) is a powerful technique for adjusting membership functions to minimize the output error. The ANFIS [38] takes into account selected input/output data to build a FIS model, where the membership functions are adjusted (tuned) either by using a backpropagation algorithm alone or in combination with a least squares-type method. This tuning allows fuzzy systems to gain knowledge from the data they are modeling. The ANFIS considers only Sugeno–Takagi–Kang [39] models that should contain a single output variable. The ANFIS is integrated into the structure of adaptive networks [40] and uses the advantages of fuzzy logic and neural networks.

### 5.1. ANFIS Structure

Our ANFIS, called ANFISBLAST, comprises different mathematical models $f$ for $f \in F$, where $F$ = {*PPV*, *FREQUENCY*}—i.e., two prediction models—as follows:

1.  *PPV*: The ANFIS model for peak particle velocity;
2.  *FREQUENCY*: The ANFIS model for frequency calculation.

The structure of the $f$ model, for each $f \in F$, is presented in Figure 6. The nodes on the left side represent the input variables and the node on the right represents the output. Each model contains two input variables, namely the charge $Q$ (kg) and the distance $d$ (m) from the point of measurement to the center of gravity of the blast, as well as a single output. In this way, two ANFIS models predict two different outputs, such as the peak particle velocity (mm/s) and the frequency (Hz).

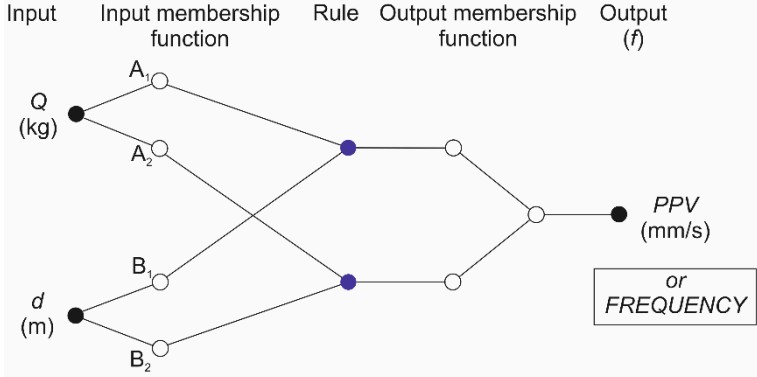

**Figure 6.** The fuzzy inference system included in the structure of adaptive network of the ANFIS-BLAST.

The structure of both models is the same, but they were constructed on the basis of different input data. For a Sugeno-type fuzzy model [39], a set of rules with $i$, $i \in I$, $I = \{1, 2\}$, and fuzzy "if–then" rules, is specified by Equations (5) and (6):

$$1. \ If \ Q \ is \ A_1 \ and \ d \ is \ B_1 \ then \ f_1 = a_0^{1,f} + a_1^{1,f} \cdot Q + a_2^{1,f} \cdot d \tag{5}$$

$$2. \ If \ Q \ is \ A_2 \ and \ d \ is \ B_2 \ then \ f_2 = a_0^{2,f} + a_1^{2,f} \cdot Q + a_2^{2,f} \cdot d \tag{6}$$

where $a_0^{i,f}$, $a_1^{i,f}$, and $a_2^{i,f}$ are the consequent parameters, and $Q$ and $d$ are the input variables. For each $f \in F$, for the $f$ model, different parameters were calculated. The general computation procedure of the ANFIS models is presented below [41]:

1. The degree of membership of a fuzzy set $(A_i, B_i)$ is computed;
2. The product of membership function for every rule is computed;
3. The proportion between the $i$th rule's strength and the sum of all of the rules' strengths is computed;
4. The output of every rule is computed;
5. The weighted average of every rule's output is computed.

In the first step, the degree of membership of the fuzzy set $(A_i, B_i)$ was computed using Equations (7) and (8):

$$\mu_{A_i}^f(Q) = exp\left[-\left(\frac{Q - c_{A_i}^f}{\sqrt{2} \cdot \sigma_{A_i}^f}\right)^2\right] \tag{7}$$

$$\mu_{B_i}^f(d) = exp\left[-\left(\frac{d - c_{B_i}^f}{\sqrt{2} \cdot \sigma_{B_i}^f}\right)^2\right] \tag{8}$$

where $Q$ and $d$ are the inputs to membership functions (Gaussian), and parameters $c_{A_i}^f$, $c_{B_i}^f$, $\sigma_{A_i}^f$, and $\sigma_{B_i}^f$ are the premise parameters. The products of the membership functions for each rule were also computed:

$$w_1^f = \mu_{A_1}^f(Q) \cdot \mu_{B_1}^f(d) \tag{9}$$

$$w_2^f = \mu_{A_2}^f(Q) \cdot \mu_{B_2}^f(d) \tag{10}$$

where $w_i$ represents the strength of the $i$th rule. The weighted average of every rule's output $\overline{w_i^f}$ was determined as a quantitative relation between the $i$th rule's strength and the sum of all the rules' strengths (see Equation (11)):

$$\overline{w_i^f} = \frac{w_i^f}{w_1^f + w_2^f} \tag{11}$$

The output of every rule was then defined as a sum of the products among the weighted average of every rule's output and a linear function between the consequent parameters and the input variables:

$$\sum_{i=1}^{2} \overline{w_i^f} \cdot \left(a_0^{i,f} + a_1^{i,f} \cdot Q + a_2^{i,f} \cdot d\right) \tag{12}$$

In this way, the same procedure was repeated for both $f$ models, $f \in F$, in order to predict the *PPV* and *FREQUENCY* (see Equations (13) and (14)). For each $f$, for the $f$ model,

the different values of the consequent parameters, premise parameters, weighted averages, and strengths of the rule outputs were evaluated.

$$PPV : PPV = \sum_{i=1}^{n} \overline{w_i^{PPV}} \cdot \left( a_0^{i,PPV} + a_1^{i,PPV} \cdot Q + a_2^{i,PPV} \cdot d \right) \tag{13}$$

$$FREQUENCY : FREQ = \sum_{i=1}^{n} \overline{w_i^{FREQ}} \cdot \left( a_0^{i,FREQ} + a_1^{i,FREQ} \cdot Q + a_2^{i,FREQ} \cdot d \right) \tag{14}$$

### 5.2. Dataset and the Modeling of the ANFISBLAST Prediction System

The ANFISBLAST prediction system was constructed to be applied to wide ranges of the charge $Q$ (kg) and distance $d$ (m) from the point of measurement to the center of gravity of the blast (see Table 6). In order to predict the *PPV* and *FREQUENCY*, the 40 events measured with a triaxial geophone were used to set the dataset for developing the ANFISBLAST prediction system. The same training data were defined for all $f$, where $f \in F$, prediction models: *PPV* and *FREQUENCY*.

**Table 6.** Limit values of each input variable.

| Input Variable | $Q$ (kg) | $d$ (m) |
|---|---|---|
| Min. value | 32 | 32 |
| Max. value | 63 | 108 |

In a basic FIS, the integer number of rules is determined by an engineer/researcher who is acquainted with the engineering problem to be modeled. There are no straightforward procedures to determine the lowest possible number of membership functions in order to obtain a required level of performance. In this paper, the number of membership functions determined for each input variable was chosen systematically by analyzing the input–output data and by using the trial-and-error method. For each $f \in F$, for the $f$ prediction model, we chose two membership functions in each input. Figure 7 shows the membership functions for $Q$ and $d$ for the prediction model *PPV*. All the integrated membership functions were Gaussian, expressed by Equations (7) and (8). Several researchers reported that Gaussian membership functions performed best among all given membership functions [42,43].

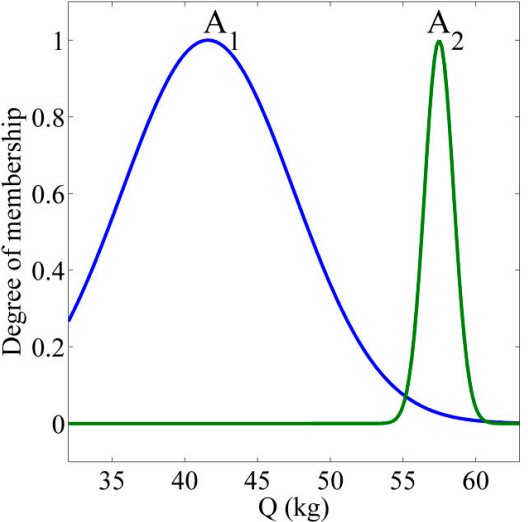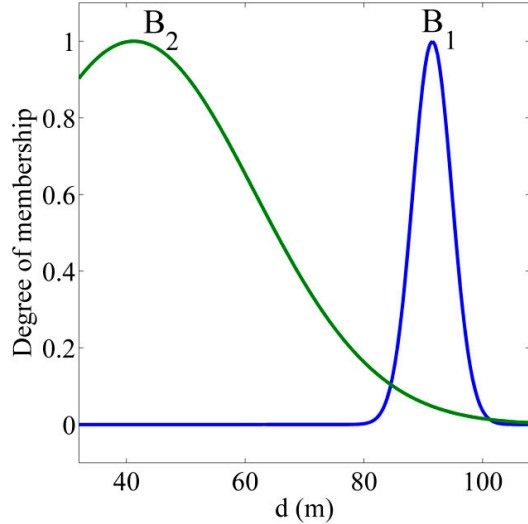

**Figure 7.** Membership functions for the prediction model peak particle velocity (PPV) for input variable $Q$, and for $d$.

Once the numbers of the membership functions related to every input were established, the initial values of the premise parameters were determined in such a way that the membership functions were uniformly distributed along the range of each input variable. For each $f \in F$, the $f$ prediction model contained two rules with two membership functions being determined for each input variable. The total quantity of fitting parameters was 10 (four premise and six consequent parameters). The abovementioned parameters were fitted by applying a hybrid algorithm. As an interface for computational modeling, MATLAB [44] was used. The premise parameters are presented in Table 7 and the consequent parameters in Table 8. The same statistical parameters as for the conventional predictors were used to determine the goodness of fit of the ANFISBLAST model for the training data. The statistical parameter values were SSE = 83.05 mm/s, R-squared = 0.73, adjusted R-squared = 0.73, and RMSE = 1.49 mm/s.

**Table 7.** Premise parameters of the *PPV* and *FREQUENCY* models.

| $n$ | *PPV* | | *FREQUENCY* | |
|---|---|---|---|---|
| | $\sigma_n$ | $c_n$ | $\sigma_n$ | $c_n$ |
| $A_1$ | 5.899 | 41.594 | 7.534 | 46.710 |
| $A_2$ | 0.998 | 57.471 | 0.459 | 50.039 |
| $B_1$ | 3.284 | 91.519 | 7.357 | 63.844 |
| $B_2$ | 20.371 | 41.201 | 12.166 | 94.635 |

**Table 8.** Consequent parameters of the *PPV* and *FREQUENCY* models.

| $i$ | *PPV* | | | *FREQUENCY* | | |
|---|---|---|---|---|---|---|
| | $a_0^i$ | $a_0^j$ | $a_2^i$ | $a_0^i$ | $a_0^j$ | $a_2^i$ |
| 1 | 11.6401 | 0.0179 | −0.1016 | 2.5956 | −0.0083 | 0.9740 |
| 2 | −3.0130 | 0.3535 | −0.1469 | 1351.6580 | −28.3040 | 1.3909 |

## 6. Testing the Conventional Vibration Predictor and ANFISBLAST Models

While training datasets were used to discover potentially predictive relationships between input variables and an output, a testing dataset was employed to check whether the prediction models had sufficient prediction compatibility. In order to test and check the conventional and ANFIS prediction models, eight testing datasets obtained from additional measurements with geophones were chosen and employed. The SSE, R-squared, adjusted R-squared, and RMSE between the predicted and measured values were taken as the measure of performance.

Table 9 shows the R-squared, SSE, RMSE, and adjusted R-squared values for all the conventional prediction models and the ANFISBLAST prediction model. From Table 9, several conclusions can be made. It was found that the performance of the developed ANFIS predictor model is superior as its SSE and RMSE are the lowest and its R-squared is the highest compared to the other conventional predictors. The average RMSE for the ANFIS model was 0.88 mm/s, whereas for the Langeforts–Kihlstorm, USBM, and Ambraseys–Hendron predictors, it was 1.49, 1.54, and 1.55 mm/s, respectively. The same conclusion can be drawn for SSE. The R-squared value was found to be highest for the developed ANFIS model. The R-squared averages for the ANFIS, Langeforts–Kihlstorm, USBM, and Ambraseys–Hendron prediction models were 0.87, 0.62, 0.59, and 0.59, respectively. The performance calculations also confirm the fact that the Langeforts–Kihlstorm model produced better results than the other two conventional predictors (i.e., USBM and Ambraseys–Hendron). Therefore, it can be concluded that the prediction capability of the ANFISBLAST system outperforms that of the conventional predictors for presented blasting site.

**Table 9.** Measure of performance for the adaptive network-based fuzzy inference system (ANFIS) and conventional predictors.

| Measurement No. | Measured (mm/s) | USBM | Ambraseys–Hendron | Langeforts–Kihlstorm | ANFISBLAST-PPV |
|---|---|---|---|---|---|
| 1 | 2.2 | 2.67 | 2.82 | 2.13 | 1.66 |
| 2 | 7.3 | 4.69 | 4.91 | 3.73 | 5.88 |
| 3 | 2.8 | 2.89 | 2.96 | 2.71 | 1.58 |
| 4 | 4.9 | 4.49 | 4.58 | 4.12 | 5.18 |
| 5 | 5.5 | 5.66 | 5.75 | 5.24 | 6.53 |
| 6 | 7.9 | 10.71 | 10.88 | 9.40 | 8.55 |
| 7 | 3.7 | 4.00 | 3.99 | 4.12 | 3.94 |
| 8 | 7.7 | 5.78 | 5.66 | 6.39 | 6.82 |
| Statistical parameter | | | | | |
| SSE | | 14.27 | 14.50 | 13.32 | 4.67 |
| R-squared | | 0.59 | 0.59 | 0.62 | 0.87 |
| Adjusted R-squared | | 0.53 | 0.52 | 0.56 | 0.84 |
| RMSE | | 1.54 | 1.55 | 1.49 | 0.88 |

## 7. Numerical Example

In order to explain the developed ANFIS technique, we have presented a calculation method for predicting the *PPV* and *FREQUENCY* caused by tunnel blasting in Pluska–Ponikve [45]. The proposed system ANFISBLAST was applied. In the numerical example, the *PPV* and *FREQUENCY* were predicted for a charge of 36 kg and a distance of 63 m. Because the proposed ANFISBLAST system comprises two different ANFIS models, two different model calculations were executed in order to predict the *PPV* and *FREQUENCY*. The predicted peak particle velocity, calculated by the ANFIS model *PPV*, yielded 5.882 mm/s. The predicted frequency, calculated by the ANFIS model *FREQUENCY*, yielded 63.661 Hz. The premise parameters for this solution are shown in Table 10 and the consequent parameters, the weighted averages of every rule's output, and the definitive output are given in Table 11.

**Table 10.** Premise parameters of the Gaussian membership function for the ANFIS models *PPV* and *FREQUENCY*.

| | ANFISBLAST-PPV | | | |
|---|---|---|---|---|
| Membership Function | Premise Parameters | | $\mu$ (Q) | $\mu$ (d) |
| $i$ | $\sigma_i^{PPV}$ | $c_i^{PPV}$ | $\mu$ (36) | $\mu$ (63) |
| $A_1$ | 7.064 | 37.168 | 0.638 | - |
| $A_2$ | 14.451 | 68.800 | 0.000 | - |
| $B_1$ | 0.870 | 2.031 | - | 0.000 |
| $B_2$ | 0.815 | 4.039 | - | 0.564 |
| | ANFISBLAST-FREQUENCY | | | |
| Membership function | Premise parameters | | $\mu$ (Q) | $\mu$ (d) |
| $i$ | $\sigma_i^{FREQ}$ | $c_i^{FREQ}$ | $\mu$ (36) | $\mu$ (63) |
| $A_1$ | 65.821 | 25.857 | 0.364 | |
| $A_2$ | 64.280 | 76.438 | 0.000 | |
| $B_1$ | 1.690 | 1.623 | | 0.993 |
| $B_2$ | 1.349 | 4.631 | | 0.034 |

**Table 11.** Consequent parameters and final outputs for the ANFIS models *PPV* and *FREQUENCY*.

| $i$ | $a_0^{i,PPV}$ | $a_1^{i,PPV}$ | $a_2^{i,PPV}$ | $w_i^{PPV}$ | $\overline{w_i^{PPV}}$ | $PPV_i = a_0^{i,PPV} + a_1^{i,PPV} \cdot Q + a_2^{i,PPV} \cdot d$ | $\overline{w_i^{PPV}} \cdot PPV_i$ |
|---|---|---|---|---|---|---|---|
| 1 | 11.6401 | 0.0179 | –0.1016 | 0.000 | 1.000 | 5.882 | 5.882 |
| 2 | –3.0130 | 0.3535 | –0.1469 | 0.000 | 0.000 | 0.459 | 0.000 |
| | | | | | | *PPV* (mm/s) | Σ 5.882 |
| $i$ | $a_0^{i,FREQ}$ | $a_1^{i,FREQ}$ | $a_2^{i,FREQ}$ | $w_i^{PPV}$ | $\overline{w_i^{FREQ}}$ | $FREQ_i = a_0^{i,FREQ} + a_1^{i,FREQ} \cdot Q + a_2^{i,FREQ} \cdot d$ | $\overline{w_i^{FREQ}} \cdot FREQ_i$ |
| 1 | 2.5956 | –0.0083 | 0.9740 | 0.362 | 1.000 | 63.661 | 63.661 |
| 2 | 1351.65 | –28.304 | 1.3909 | 0.000 | 0.000 | 420.342 | 0.000 |
| | | | | | *FREQUENCY* (Hz) | | Σ 63.661 |

## 8. Conclusions

This paper addressed the application of an ANFIS as a blast-induced vibration predictor. Based on the research papers reviewed, this technique has not yet been employed for blast-induced ground vibration and frequency prediction. In order to confirm the superiority of the ANFIS, the ANFISBLAST prediction model was tested and compared with the three most widely used conventional predictors. The predictors were evaluated in terms of their SSE, RMSE, R-squared, and adjusted R-squared values. Since the ANFIS can detect patterns in training datasets and can be updated when new training datasets are presented, a better degree of accuracy can be achieved compared to other conventional techniques. Considering the complex relationship between the inputs and outputs, the achieved results were satisfactory. However, it should be noted that the ANFISBLAST model is not directly applicable to the vibration analysis on other tunnel construction sites. Further prediction methods could be investigated in future studies based on the data provided in this article, and the most robust and accurate model should be used to predict ground vibration levels.

**Author Contributions:** Conceptualization, P.J., S.L., and A.I.; methodology, P.J. and S.L.; formal analysis, P.J., S.L., and A.I.; writing—original draft preparation, P.J., S.L., and A.I. All authors have read and agreed to the published version of the manuscript.

**Funding:** This research was funded by Slovenian Research Agency, grant number P2-0268.

**Institutional Review Board Statement:** Not applicable.

**Informed Consent Statement:** Not applicable.

**Data Availability Statement:** Data sharing not applicable.

**Conflicts of Interest:** The authors declare no conflict of interest.

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
