# Peer review of "Prediction of Blast-Induced Ground Vibration Using an Adaptive Network-Based Fuzzy Inference System"

_applsci, doi:10.3390/app11010203_

Round 1

Reviewer 1 Report

The paper discusses the prediction of blast-induced ground vibrations using ANFIS. The authors have used very little input parameters and little data (40) for prediction. See more comments in the attached file. 

Author Response

List of actions, changes and comments

Prediction of blast-induced ground vibration using an adaptive network-based fuzzy inference system

APPLIED SCIENCES

by Primož Jelušič, Andrej Ivanič and Samo Lubej

Actions, Changes and Comments:

Reviewer 1:

1) Why ANFIS?  Why not anything else?

The ANFIS combines the advantages of ANN and Fuzzy Inference Systems (FIS). Thus, it has rapid learning capacity, the capability of seizing the nonlinear structure of a process and the capability of adaptation.

The sentence was improved to:

“Efforts were made to predict and evaluate blast-induced ground vibrations and frequencies using an adaptive network-based fuzzy inference system (ANFIS), which has a fast-learning capability and the ability to capture the non-linear response during the blasting process.”

2) Cite all the facts quoted from published literature. If not published then provide basis or results

A new text on the published research work has been added as manuscript.

Khandelwal and Singh [2] used ANN to predict the blast-induced ground vibration level at a magnesite mine based on 75 blast events. Later, Khandelwal and Singh [3] developed an ANN model for predicting ground vibrations and frequencies that considers not only the distance from the blast site and the charge per delay, but also rock properties, blast design and explosive parameters. The sensitivity analysis of different types of ANN models showed that the distance from the blast site, the number of boreholes per delay and the maximum charge per delay are the most effective parameters in generating ground vibrations during blasting [4, 5]. Kostič et al. developed a neural network model with four main blast parameters as input, namely total charge, maximum charge per delay, distance from the blast source to the measuring point and hole depth [6]. To evaluate the ground vibrations caused by the blast, the techniques of dimensional analysis and ANN were applied, taking into account the blast design parameters and rock strength [7]. The control of the blast induced vibrations during the construction of the Masjed-Soleiman dam was the crucial task, therefore the general regression neural network (GRNN) and the support vector machine (SVM) were used to predict the vibrations [8]. A hybrid model of the ANN and a particle swarm optimization (PSO) algorithm were implemented to predict ground vibrations based on 88 blast events [9]. Two novel hybrid artificial intelligent models for predicting the blast-induced peak particle velocity were presented by Li, et al. [10]. Rao and Rao [11] applied the neuro-fuzzy technique for ground vibration and frequency prediction in opencast mine while Khendelwal [12] applied the ANN. Suchatvee et. al. [13] investigated the advantages and limitations of ANNs for the prediction of surface settlements generated by earth pressure balance shield tunneling. The importance of site-specific factors in the prediction model for blast-induced ground vibrations was presented by Kuzu [14]. ANNs have also been used to analyze the surface settlements caused by shotcrete-supported tunnels [15] and to predict Tunnel Boring Machine Penetration Rate [16]. The summary of the previous studies on the PPV, which include several soft computing and machine learning methods, was presented by Zhang et al. [17].

3) The authors should read more papers on blast induced vibrations

New references have been added and the text has been improved by citing recent research.

4) Charge weight scaling law

The charge weight scaling law was added to the manuscript. The new sentence was added. 

The relationship between the peak level of the vibration, the distance from the source to the monitor and the total charge weight in the blast hole is defined by the charge weight scaling law [1].

  1. Blair, D.P. Blast vibration dependence on charge length, velocity of detonation and layered media. Int. J. Rock Mech. Min. Sci. 2014, 65, 29–39, doi:10.1016/j.ijrmms.2013.11.007.

5) PPV

It was corrected.

Permissible vibration was changes to peak particle velocity.

6) BV monitoring is not a tool but a component

The sentence was corrected. The word tool was replaced with word component.

7) what does technical results mean?

The vague term “technical results” was deleted.

8) There are too many variables. It is not realistic to predict blast vibrations based on only two inputs. These are not enough.  Add more inputs like rock type, explosive type, initiation method

The number of variables in the ANFIS model is the same as in the simplified equations. The type of rock, the type of explosive and initiation method could not be considered in the ANFIS model, while these were not changed during the blasting.

9) these are older methods

These are older methods but are often used for comparison with the ANN model.

10) Most of the information in section 2 is very basic. I think the audience should have this.

Additional information was added to the text describing the correlation between wave velocity and geotechnical properties of the rock. The new sentences were added:

Several studies have related pressure wave velocity (Vp) with mechanical and physical properties such as stiffness, strength and density of rock mass [23]. The Vp for elastic and isotropic media is calculated with Eq. (1) [24]:

(1)

where Kb is the bulk modulus, G is the shear modulus and ρ is the bulk density. The summary of relationship between pressure wave velocity with density and uniaxial compressive strength was presented by Yagiz [25].

 11) These properties are for intact rock and not rock mass... In reality there is no intact rock on site. Only rock mass exists.

The text was improved. The following sentences were added:

However, these properties apply to intact rock, which must be reduced due to the properties of joints in a rock mass. Therefore, several empirical equations have been developed to estimate the value of an isotropic rock mass deformation modulus. The most important classification systems for the characterization of rock mass are the Geological Strength Index (GSI) [20], the Rock Mass Rating (RMR) [21] and the Tunneling Quality Index (Q) [22]

The references were added:

  1. Hoek, E.; Brown, E.T. Practical estimates of rock mass strength. Int. J. Rock Mech. Min. Sci. 1997, 34, 1165–1186, doi:10.1016/S1365-1609(97)80069-X.
  2. Bieniawski, Z.T. Engineering classification of rock masses. Trans. S. African. Inst. Civ. Engrs. 1973, 15, 335‑344.
  3. Barton, N.; Lien, R.; Lunde, J. Engineering classification of rock masses for the design of tunnel support. Rock Mech. 1974, 6, 189–236, doi:10.1007/BF01239496.

12) these are not the only parameters

The sentence was improved.

The main controllable parameters are the type and quantity of the explosive, sequence of initiation, powder factor, drilling, steaming and hole depth while the main uncontrollable parameters are rock mass properties, geology, and joint formation.

13) there are more parameters than the ones present here.

The sentence was improved.

The main controllable parameters are the type and quantity of the explosive, sequence of initiation, powder factor, drilling, steaming and hole depth while the main uncontrollable parameters are rock mass properties, geology, and joint formation.

14) inversely proportional

The sentence was corrected. The word inversely proportional is used.

15) spell check

It was corrected to charge per delay.

16) How to find rock constant?

The on-site constants (K, B) can be obtained by using multiple regression analysis.

17) How to find these constants?

The following text was added with the reference.

To determine the damping coefficient, the measured attenuation data should be well matched with the predicted data [32].

The reference was added:

  1. Kim, D.; Lee, J. Propagation and attenuation characteristics of various ground vibrations. 2000, 19, 115–126.

18) Data scattering why this happens?

The following text was added with the reference.

Agrawal and Mishra [33] reported that the errors between predicted and actual PPV are due to the fact that cap scattering in pyrotechnic based delay initiation system varies between ±10% and ±20%.

The reference was added:

  1. Agrawal, H.; Mishra, A.K. Probabilistic analysis on scattering effect of initiation systems and concept of modified charge per delay for prediction of blast induced ground vibrations. Measurement 2018, 130, 306–317, doi:https://doi.org/10.1016/j.measurement.2018.08.032.

19) “Ambraseys–Hendron” double check this

It has been corrected. The square root has been removed from the equation. The equation now is:

20) old data?

The blasting in two tunnel tubes was carried out from March to the end of July 2008. However, the ground vibration data have still not been analyzed and published.

21) which one is critical? Why magnitude of vertical is greatest?

The axis of the geophone was aligned with the direction of blast source. In this blast event, the vertical component of the vibration is highest while the blast source is below the geophone.

22) How PPV affects the structure?

The text describing the crack width measurement on the building was added and a new figure (Figure 4) was added.

Before the blasting of the tunnel tubes, the 7 cracks were found on the outer walls of the building. We mapped the cracks and measured their width. The movements or displacements of these cracks were observed with installed plaster seals. During the blasting, it was revealed that the cracks did not enlarge and no new cracks formed.

23) What is the effect of frequency?

The following text was added with additional reference.

The deviations of the fundamental natural frequencies can influence the structural strength of the building. However, for failure mechanisms such as bending and shear, where the building is forced to follow the oscillatory movements of the ground surface, the deviation of the fundamental natural frequencies of the building is of minor importance [31].

  1. Norén-Cosgriff, K.M.; Ramstad, N.; Neby, A.; Madshus, C. Building damage due to vibration from rock blasting. Soil Dyn. Earthq. Eng. 2020, 138, 106331, doi:10.1016/j.soildyn.2020.106331.

24) Why data is inconsistent?

Such data was collected during the monitoring.

25) How many repetitions were performed for each measurement?

The following text was added in the manuscript.

Continuous record mode was used to record multiple events automatically with no dead time between blast events. The geophone records a blast event and then continues to monitor, ready to record the following events. The geophone records all blast events with a PPV exceeding the 0.2 mm/s.

26) “Squares method” Why?

The text was added.

The least squares method is often used to generate estimators and other statistics in regression analysis.

27) why USBM predictor?

The following text was added.

The USBM [34] predictor is used while it provides the safe level blasting criteria, although it is used to predict blast induced ground vibrations from surface mining rather than tunnel blasting.

28) How to compute A1 A2 A3 and A4 in ANFISBLAST

A1, A2, B1 and B2 represent the membership functions in the ANFISBLAST model. These membership functions are defined with two parameters σn and cn. These parameters are adjusted to minimize the difference between measured and predicted PPV. This was achieved by a backpropagation algorithm and a least squares method (hybrid algorithm).

29) 40 events are enough for developing ANFIS model?

ANFIS is a kind of ANN method. For vibration prediction, other authors (e.g. Saadat, M.; Khandelwal, M.; Monjezi, M. An) have used 40 training data sets to create ANN models. However, more training data leads to the ability to better predict PPV.

Saadat, M.; Khandelwal, M.; Monjezi, M. An ANN-based approach to predict blast-induced ground vibration of Gol-E-Gohar iron ore mine, Iran. J. Rock Mech. Geotech. Eng. 2014, 6, 67–76, doi:10.1016/j.jrmge.2013.11.001.

30) What are membership functions as written in line 250?

Membership functions are functions that return the degree of membership, how a crisp value is mapped to an input space. They don’t have any physical meaning.

31) Why gaussain function is obtained? What is the rational?

The following text was added with references.

Several researchers reported that Gaussian membership functions performed best among all given membership functions [42, 43].

32) Statistically it looks good, but we are not sure and we cannot say that it outperforms unless we try and test it on a few other blast sites

The text was corrected:

Therefore, it can be concluded that the prediction capability of the ANFISBLAST system outperforms that of the conventional predictors for presented blasting site.

33) not a large number

This sentence was deleted.

34) not very difficult

This sentence was deleted.

35) read more papers (recent) and add in the review

The following recent research paper were added among others:

Zhang, H.; Zhou, J.; Jahed Armaghani, D.; Tahir, M.M.; Pham, B.T.; Huynh, V. Van A Combination of Feature Selection and Random Forest Techniques to Solve a Problem Related to Blast-Induced Ground Vibration. Appl. Sci. 2020, 10, 869, doi:10.3390/app10030869.

Li, G.; Kumar, D.; Samui, P.; Nikafshan Rad, H.; Roy, B.; Hasanipanah, M. Developing a New Computational Intelligence Approach for Approximating the Blast-Induced Ground Vibration. Appl. Sci. 2020, 10, 434, doi:10.3390/app10020434.

Norén-Cosgriff, K.M.; Ramstad, N.; Neby, A.; Madshus, C. Building damage due to vibration from rock blasting. Soil Dyn. Earthq. Eng. 2020, 138, 106331, doi:10.1016/j.soildyn.2020.106331.

Xu, H.; Zhou, J.; G. Asteris, P.; Jahed Armaghani, D.; Tahir, M.M. Supervised Machine Learning Techniques to the Prediction of Tunnel Boring Machine Penetration Rate. Appl. Sci. 2019, 9, 3715, doi:10.3390/app9183715.

Agrawal, H.; Mishra, A.K. Probabilistic analysis on scattering effect of initiation systems and concept of modified charge per delay for prediction of blast induced ground vibrations. Measurement 2018, 130, 306–317, doi:10.1016/j.measurement.2018.08.032.

Hajihassani, M.; Jahed Armaghani, D.; Monjezi, M.; Mohamad, E.T.; Marto, A. Blast-induced air and ground vibration prediction: a particle swarm optimization-based artificial neural network approach. Environ. Earth Sci. 2015, 74, 2799–2817, doi:10.1007/s12665-015-4274-1.

We are grateful for reviewer comments.

Primož Jelušič, Andrej Ivanič and Samo Lubej

Reviewer 2 Report

Dear Authors

Please refer to my comments listed below. Most of them are not mandatory (I marked minor revision), however I believe, if considered, may improve the value of your article.  

  1. line 39 - please avoid multiple references like [1-4], refer separately to each paper - that will help the reader to understand why is this particular paper important for your study.
  2. line 62 - when describing types of seismic waves, please try to explain the range of their importance in transmitting energy through the soil/rock body.
  3. line 70 - provide source of information in Table 1. Is wave velocity in marble really higher than in granite? Please present a formulae describing dependence between wave velocity, stiffness (modulus) and density of the rock.
  4. line 89 - please refer to code/guideline/recommendation USBM RI 8507. I'd appreciate some more information (kind of state of the art) about codes, guidelines, and recommendations for various building structures in course of shokcs 
  5. line 94 - please provide source of Figure 1. 
  6. line 131 - please provide an arrow with distance from the bilding to blasting works on Figure 2
  7. line 133 - please do not hide microphone on Figure 3. It looks a little bit funny for a person who uses Minimate for everyday :-)
  8. line 135 - please provide more detailed information on measurements: sampling frequency, time periods ...
  9. line 146 - please explain what does "transverse" and "longitudinal" means in this particular case
  10. lines 331-335. I'd strongly oppose your conclusion about direct applicability of proposed method to the vibration analysis in other destinations. It would be fair to say that "proposed methodology" may be applied in other cases, but not directly currently obtained results. 

Author Response

Prediction of blast-induced ground vibration using an adaptive network-based fuzzy inference system

APPLIED SCIENCES

by Primož Jelušič, Andrej Ivanič and Samo Lubej

Actions, Changes and Comments:

Reviewer 2:

  1. line 39 - please avoid multiple references like [1-4], refer separately to each paper - that will help the reader to understand why is this particular paper important for your study.

Multiple references have been avoided and new, more recent references have been added. A new text on the published research work has been added in manuscript.

Khandelwal and Singh [2] used ANN to predict the blast-induced ground vibration level at a magnesite mine based on 75 blast events. Later, Khandelwal and Singh [3] developed an ANN model for predicting ground vibrations and frequencies that considers not only the distance from the blast site and the charge per delay, but also rock properties, blast design and explosive parameters. The sensitivity analysis of different types of ANN models showed that the distance from the blast site, the number of boreholes per delay and the maximum charge per delay are the most effective parameters in generating ground vibrations during blasting [4, 5]. Kostič et al. developed a neural network model with four main blast parameters as input, namely total charge, maximum charge per delay, distance from the blast source to the measuring point and hole depth [6]. To evaluate the ground vibrations caused by the blast, the techniques of dimensional analysis and ANN were applied, taking into account the blast design parameters and rock strength [7]. The control of the blast induced vibrations during the construction of the Masjed-Soleiman dam was the crucial task, therefore the general regression neural network (GRNN) and the support vector machine (SVM) were used to predict the vibrations [8]. A hybrid model of the ANN and a particle swarm optimization (PSO) algorithm were implemented to predict ground vibrations based on 88 blast events [9]. Two novel hybrid artificial intelligent models for predicting the blast-induced peak particle velocity were presented by Li, et al. [10]. Rao and Rao [11] applied the neuro-fuzzy technique for ground vibration and frequency prediction in opencast mine while Khendelwal [12] applied the ANN. Suchatvee et. al. [13] investigated the advantages and limitations of ANNs for the prediction of surface settlements generated by earth pressure balance shield tunneling. The importance of site-specific factors in the prediction model for blast-induced ground vibrations was presented by Kuzu [14]. ANNs have also been used to analyze the surface settlements caused by shotcrete-supported tunnels [15] and to predict Tunnel Boring Machine Penetration Rate [16]. The summary of the previous studies on the PPV, which include several soft computing and machine learning methods, was presented by Zhang et al. [17].

  1. line 62 - when describing types of seismic waves, please try to explain the range of their importance in transmitting energy through the soil/rock body.

The following text was added:

In rock masses, pressure waves propagate both through the mineral structure and through the pores, which is why the pressure wave velocity is increased in a saturated hard rock. Shear waves oscillating perpendicular to the wave direction propagate only through the mineral structure, therefore water saturation only has a small influence on the velocity of the shear waves. The energy of shear waves is less easily transmitted through rock mass in comparison to the energy of primary waves.

  1. line 70 - provide source of information in Table 1. Is wave velocity in marble really higher than in granite? Please present a formulae describing dependence between wave velocity, stiffness (modulus) and density of the rock.

The reference has been added to Table 1. The equation describing the dependence between wave velocity, modulus and density of the rock was added in the text. New references that summarize the relationship between wave velocity and geotechnical parameters of the rock. The following text was added:

However, these properties apply to intact rock, which must be reduced due to the properties of joints in a rock mass. Therefore, several empirical equations have been developed to estimate the value of an isotropic rock mass deformation modulus. The most important classification systems for the characterization of rock mass are the Geological Strength Index (GSI) [20], the Rock Mass Rating (RMR) [21] and the Tunneling Quality Index (Q) [22]. Several studies have related pressure wave velocity (Vp) with mechanical and physical properties such as stiffness, strength and density of rock mass [23]. The Vp for elastic and isotropic media is calculated with Eq. (1) [24]:

where Kb is the bulk modulus, G is the shear modulus and ρ is the bulk density. The summary of relationship between pressure wave velocity with density and uniaxial compressive strength was presented by Yagiz [25].

  1. line 89 - please refer to code/guideline/recommendation USBM RI 8507. I'd appreciate some more information (kind of state of the art) about codes, guidelines, and recommendations for various building structures in course of shokcs.

The word recommendation was replaced with code/guideline/recommendations.

  1. line 94 - please provide source of Figure 1. 

The reference has been added to Figure 1.

The peak particle velocity (m/s) is on the vertical scale and the vibration frequency (Hz) is on the horizontal scale, as presented in Figure 1 [28].

  1. line 131 - please provide an arrow with distance from the building to blasting works on Figure 2.

The distance was added on Figure 2.

  1. line 133 - please do not hide microphone on Figure 3. It looks a little bit funny for a person who uses Minimate for everyday :-)

The figure has been corrected. In fact, sound monitoring was carried out during the blasting.

  1. line 135 - please provide more detailed information on measurements: sampling frequency, time periods ...

Additional information about vibration measurement has been added. The following text was added:

Continuous record mode was used to record multiple events automatically with no dead time between blast events. The geophone records a blast event and then continues to monitor, ready to record the following events. The geophone records all blast events with a PPV exceeding the 0.2 mm/s.

  1. line 146 - please explain what does "transverse" and "longitudinal" means in this particular case

The axis of the geophone was aligned with the expected vibration source.

  1. lines 331-335. I'd strongly oppose your conclusion about direct applicability of proposed method to the vibration analysis in other destinations. It would be fair to say that "proposed methodology" may be applied in other cases, but not directly currently obtained results.

The additional explanation was added.  

However, it should be noted that the ANFISBLAST model is not directly applicable to the vibration analysis on other tunnel construction sites.

We are grateful for reviewer comments.

Primož Jelušič, Andrej Ivanič and Samo Lubej

Round 2

Reviewer 1 Report

Seems like the paper has some changes. The comments have been incorporated well. You can accept the paper as is.